# Utility of a Molecular Signature for Predicting Recurrence and Progression in Non-Muscle-Invasive Bladder Cancer Patients: Comparison with the EORTC, CUETO and 2021 EAU Risk Groups

**DOI:** 10.3390/ijms232214481

**Published:** 2022-11-21

**Authors:** Xuan-Mei Piao, Seon-Kyu Kim, Young Joon Byun, Chuang-Ming Zheng, Ho Won Kang, Won Tae Kim, Yong-June Kim, Sang-Cheol Lee, Wun-Jae Kim, Sung-Kwon Moon, Yung Hyun Choi, Seok Joong Yun

**Affiliations:** 1Department of Urology, College of Medicine, Chungbuk National University, Cheongju 28644, Republic of Korea; 2Personalized Genomic Medicine Research Center, Korea Research Institute of Bioscience and Biotechnology, Daejeon 34141, Republic of Korea; 3Department of Urology, Chungbuk National University Hospital, Cheongju 28644, Republic of Korea; 4Urotech Institute, Cheongju 28120, Republic of Korea; 5Department of Food Science and Technology, Chung-Ang University, Ansung 456-756, Republic of Korea; 6Department of Biochemistry, College of Oriental Medicine, Dong-Eui University, Busan 614-052, Republic of Korea

**Keywords:** molecular subtype, risk score, recurrence, progression, non-muscle invasive bladder cancer

## Abstract

To evaluate the utility of different risk assessments in non-muscle-invasive bladder cancer (NMIBC) patients, a total of 178 NMIBC patients from Chungbuk National University Hospital (CBNUH) were enrolled, and the predictive value of the molecular signature-based subtype predictor (MSP888) and risk calculators based on clinicopathological factors (EORTC, CUETO and 2021 EAU risk scores) was compared. Of the 178 patients, 49 were newly analyzed by the RNA-sequencing, and their MSP888 subtype was evaluated. The ability of the EORTC, MSP888 and two molecular subtyping systems of bladder cancer (Lund and UROMOL subtypes) to predict progression of 460 NMIBC patients from the UROMOL project was assessed. Cox regression analyses showed that the MSP888 was an independent predictor of NMIBC progression in the CBNUH cohort (*p* = 0.043). Particularly in patients without an intravesical BCG immunotherapy, MSP888 significantly linked with risk of disease recurrence and progression (both *p* < 0.05). However, the EORTC, CUETO and 2021 EAU risk scores showed disappointing results with respect to estimating the NMIBC prognosis. In the UROMOL cohort, the MSP888, Lund and UROMOL subtypes demonstrated a similar capacity to predict NMIBC progression (all *p* < 0.05). Conclusively, the MSP888 is favorable for stratifying patients to facilitate optimal treatment.

## 1. Introduction

In 2022, an estimated 60,800 adults will be newly diagnosed with non-muscle invasive bladder cancer (NMIBC) in the United States, which is approximately 75% of all bladder cancer (BCa) patients, and in younger patients (<40 years of age) this percentage is even higher [1]. NMIBC is one of the costliest cancers to treat due to the heterogeneity of the disease and high recurrence rates; this means that patients must be monitored throughout their life [2]. Treatment of NMIBC aims mainly to reduce recurrence and to prevent progression to a muscle-invasive phenotype [3]. Therefore, predicting recurrence and progression in individual patients is crucial to facilitate optimal treatment and management protocols.

Risk tables such as the European Organization for Research and Treatment of Cancer (EORTC) and the Spanish Urology Association for Oncological Treatment (CUETO) can be used to predict NMIBC recurrence and progression [4]. The EORTC calculates a risk score based on the number of tumors, tumor size, prior recurrence, T stage, presence of carcinoma in situ (CIS) and grade (1997 WHO grading system) [5]. The main limitation of the EORTC model is a lack of patients treated with Bacillus Calmette-Guérin (BCG). More recently, CUETO proposed a modified model that predicts the probability of disease recurrence and progression in BCG-treated patients [6]. CUETO includes fewer factors (age, gender, tumor recurrence, number of tumors, T stage, CIS and grade (1997 WHO grading system)) than EORTC [6]; however, external validation studies performed in different countries suggest that the applicability of EORTC and CUETO is controversial [7]. The 2021 European Association of Urology (EAU) guidelines suggested updated risk groups—which were calculated by patients’ age, tumor stage, tumor grade (both 1997 and 2004 WHO grading systems), tumor number and size, and concomitant CIS—for predicting NMIBC progression. However, this scoring model did not account for the patients treated with BCG and could only determine the risk of tumor progression but not recurrence [8].

Risk assessment for NMIBC recurrence and progression by these predictive models that are recommended by current EAU guidelines remains an unsolved issue due to the ambiguity of their performance in actual clinical practice in different regions.

The molecular landscape in NMIBC is based on broad biological heterogeneity, which leads to unexpected clinical outcomes. Much effort has been put into examining tumor heterogeneity at the molecular level, mainly by analyzing genomic alterations in NMIBC; indeed, several studies have revealed complex genomic patterns underlying bladder tumorigenesis [9,10,11,12]. In a previous study, we developed a molecular signature-based subtype predictor (called MSP888) comprising three different subtypes; MSP888 differentiates NMIBC patients with diverse prognoses and different responses to BCG therapy [13]. The goal of the present study was to confirm the utility of MSP888 as a prognostic model for NMIBC, and to compare its predictive efficacy with that of risk scoring models based on clinicopathological factors in two different study cohorts.

## 2. Results

### 2.1. Construction of the MSP888 Classifier and Validation of Its Prognostic Value in the CBNUH RNA-seq Cohort

The molecular clusters in the Chungbuk National University Hospital (CBNUH) RNA-sequencing (RNA-seq) cohort were defined based on the MSP888 classifier. As in our previous study, the optimal gene sets, comprising 888 genes, whose expression values were significantly diverse among Clusters 1, 2 and 3 were used to classify the molecular subgroups of NMIBC in the CBNUH RNA-seq cohort. A deep belief network-based (DBN) [14] deep learning method was then applied to develop and train the prediction model (Figure 1A). Appendix A shows the detailed data, along with predicted cluster results. Kaplan–Meier survival plots showing the recurrence-free survival (RFS) and progression-free survival (PFS) of NMIBC patients sub-grouped by the MSP888 prediction model were generated (Figure 1B–Figure 2E). The recurrence and progression rates of NMIBC patients in Cluster 3 were higher than those in the other clusters (Figure 1B,D); however, due to the small sample size with a low incidence rate, the *p*-Value was not significant. When we compared the RFS and PFS in individual clusters, the recurrence rate of NMIBC patients classified into Cluster 2 was dependent on BCG treatment that patients treated with BCG suffered worse RFS (*p* = 0.001) (Figure 1C).

### 2.2. MSP888 Classifier Predicts NMIBC Prognosis in the CBNUH Cohort Better Than the EORTC, CUETO and 2021 EAU Risk Groups

In the previous study, we used 129 NMIBC samples to estimate the power of MSP888 for predicting NMIBC prognosis [13]. Here, we examined 49 cases in the CBNUH RNA-seq cohort and then compared the prognostic ability of the MSP888 classifier to that of EORTC, CUETO and 2021 EAU risk groups acquired from a total of 178 NMIBC patients. The correlation between each risk estimator and the prognosis of NMIBC patients was explored using Pearson’s Chi-square test. The interval likelihood ratio (LR) was calculated as the probability of an individual test result occurring when recurrence/progression is present divided by the probability of an individual test result occurring when recurrence/progression is absent in order to demonstrate how likely it is that a patient will experience recurrence or progression

#### 2.2.1. Analysis of NMIBC Recurrence

Only the MSP888 classifier showed a significant association with recurrence in patients without BCG treatment (Pearson’s Chi-square *p* = 0.024) (Table 1). When we calculated the interval LR for recurrence in the MSP888 classifier of NMIBC patients not treated with BCG, we found that Cluster 2 had a significantly higher LR of recurrence (LR, 1.942; 95% CI, 1.257–3.000) than Clusters 1 or 3 (Table 1). However, neither the EORTC nor the CUETO risk scores correlated with NMIBC recurrence. Kaplan–Meier survival plots demonstrated that patients in Cluster 2 suffered more recurrence than those in Cluster 1 (log-rank test, *p* = 0.027) (Figure 2B). In addition, univariate and multivariate Cox regression analyses of the three clusters in the CBNUH cohort, along with known clinicopathological risk factors and the EORTC and CUETO risk scores, were conducted to determine prognostic factors. CUETO risk scores were calculated to determine prognostic factors. Univariate analysis identified tumor grade (both 1997 and 2004 WHO grading systems), the CUETO score and MSP888 as significant prognostic indicators of RFS in NMIBC patients not treated with BCG (all *p* < 0.05) (Table 2). Because the tumor grade is an index used to calculate the CUETO score, we supposed that the prognostic significance of the CUETO score may be affected by tumor grade (1997 WHO grading system); therefore, we included only the tumor grade and the MSP888 classifier in multivariate analysis. Multivariate Cox analysis revealed that the MSP888 classifier retained statistical significance for RFS of NMIBC patients (HR, 4.104; 95% CI, 1.319–12.770; *p* = 0.015) (Table 2). These results illustrate the high prognostic relevance of Cluster 2, characterized as a REC.BCG+ subtype associated with a poor prognosis for disease recurrence, but with a good response to BCG treatment.

#### 2.2.2. Analysis of NMIBC Progression

Here, we found that the MSP888 classifier was able to differentiate NMIBC patients with progression from those without (Pearson’s Chi-square, *p* < 0.0001). In addition, the interval LR suggested that patients in Cluster 3 were more likely to suffer progression than those in Clusters 1 or 2 (LR, 2.738; 95% CI, 1.863–4.024); however, none of the EORTC, CUETO and 2021 EAU risk scores correlated with NMIBC progression (Table 1). In particular, both the MSP888 classifier and the CUETO risk score showed a relevant correlation in NMIBC patients treated with BCG treatment: the LR for patients in Cluster 3 and for patients with a high CUETO risk score were 3.840 and 4.000, respectively, which is higher than for the other groups (Table 1). Kaplan–Meier survival plots revealed that the PFS of patients in Cluster 3 was poorer than that for patients in Cluster 1 or 2 (log-rank test, *p* < 0.0001, respectively) (Figure 2D and E). Univariate Cox analysis identified age, tumor grade (both 1997 and 2004 WHO grading systems), the CUETO score and MSP888 as significant indicators of PFS in NMIBC patients (Table 3). As mentioned above, the prognostic significance of the CUETO score may be largely influenced by tumor grade (1997 WHO grading system); therefore, multivariate Cox analyses were evaluated using tumor grade and the MSP888 classifier. The results showed that the MSP888 classifier was a crucial independent risk factor for NMIBC progression (HR, 4.285; 95% CI, 1.708–10.750; *p* = 0.002); this was particularly true for patients not treated with BCG, in whom the HRs of the MSP888 classifier were 14.619 (95% CI, 4.042–52.869; *p* < 0.0001) and 7.411 (95% CI, 1.985–27.669; *p* = 0.003) (Table 3). These results suggest the high prognostic value of Cluster 3, characterized as the DP.BCG+ subtype; this subtype has a poor prognosis with respect to disease progression, but a good response to BCG treatment. Taken together, the above results indicate that the MSP888 classifier is a more powerful predictor of NMIBC prognosis than the EORTC, CUETO and 2021 EAU risk scores.

### 2.3. In the UROMOL Cohort, MSP888 Classifier Predicts NMIBC Prognosis as well as Previously Developed Molecular Classifications

The UROMOL project is a European multicenter prospective study that aims to identify molecular markers that predict the likelihood of progression in patients with NMIBC [9]. The UROMOL study sub-grouped NMIBC into three major classes with basal- and luminal-like characteristics and different clinical outcomes, which is partly consistent with Lund subtypes [9]. In the current study, we compared the predictive efficacy of the MSP888 classifier with that of the UROMOL classification, the Lund subtype and the EORTC score in 460 NMIBC patients enrolled in the UROMOL project. Due to a lack of clinical information recorded by the UROMOL project, we could only assess the risk of progression. Table 4 summarizes the associations between NMIBC progression and the four risk estimators: MSP888 classifier, UROMOL classification, Lund subtype and EORTC risk score. All four risk predictors correlated with NMIBC progression, especially in patients not treated with BCG (Pearson Chi-square *p* < 0.05). The interval LR revealed that class 2 of the UROMOL classification, the genomically unstable Lund subtype, a high EORTC risk score and Cluster 3 of the MSP888 were associated with a higher risk of progression. Thus, the predictive performance of the four risk estimators for NMIBC progression is in line with that reported previously in the UROMOL and CBNUH cohort studies.

To identify and compare the prognostic independence of the MSP888 classifier, the UROMOL classification and the Lund subtype with respect to NMIBC progression, we applied Cox regression analyses to each risk predictor in the UROMOL cohort, along with known clinicopathological risk factors such as the EORTC score. Univariate Cox analysis identified age and tumor stage, along with the four risk estimators, as independent prognostic indicators of PFS in NMIBC (Table 5). Because tumor stage is an index used to calculate the EORTC, we excluded the EORTC from the following multivariate Cox analyses. Multivariate analysis based on tumor age, stage and the MSP888 classifier, or the UROMOL classification, or the Lund subtype revealed that the MSP888 classifier and the Lund subtype retained statistical significance with respect to PFS of NMIBC patients (HR, 3.025 and 3.605; 95% CI, 1.277–7.166 and 1.200–10.826; and *p* = 0.012 and 0.022, respectively) (Table 5). Kaplan–Meier analysis revealed that NMIBC patients in Cluster 3 of MSP888, in class 2 of the UROMOL classification or with a genomically unstable Lund subtype experienced more progression than patients classified as other subtypes (log-rank test, *p* < 0.05) (Figure 3), confirming the high prognostic relevance of the three clusters classified by the MSP888.

Finally, we evaluated the prognostic significance of the four risk predictors for NMIBC progression in patients not treated with BCG. Univariate and multivariate Cox regression analyses revealed that MSP888 classifier and Lund subtype were independent predictors of PFS. The risk of progression for patients in Cluster 3 of the MSP888 was 3.753 times higher than that for patients in Clusters 1 and 2 (*p* = 0.005). In addition, the progression risk for patients with a genomically unstable Lund subtype was 5.962 times higher than that of patients with other subtypes (*p* = 0.005) (Table 6). The PFS of NMIBC patients not treated with BCG was estimated from Kaplan–Meier survival plots. Patients in Cluster 3 of the MSP888, in class 2 of UROMOL or with a genomically unstable Lund subtype experienced more progression than other subgroups (log-rank test, *p* < 0.05) (Figure 4), suggesting the crucial prognostic relevance of the three clusters classified by the MSP888, which is comparable with that of previously constructed global subtypes.

## 3. Discussion

Most traditional classification systems for BCa are based on pathological parameters [15,16]. However, there are large differences between individuals with respect to prognosis of BCa, even in those with tumors of similar pathological stage and grade [17,18]. Accordingly, an omnibus risk score table (i.e., EORTC), which is based on clinicopathologic characteristics, was developed in 2006 to assess the probability of recurrence and progression in patients with NMIBC [5]. However, the lack of patients treated with BCG led to development of the CUETO risk tables to predict the risk of recurrence and progression specifically in patients treated with BCG [6]. To estimate 1-year and 5-year recurrence or progression rates, the EORTC and CUETO risk scores classify patients into four groups (low, intermediate low, intermediate high and high). Several external validation studies have been performed to evaluate the applicability of EORTC and CUETO in different countries, although the results are controversial [7]. For example, some patients were pathologically classified as low risk, but the actual tumor showed highly invasive biological characteristics and early metastasis. Therefore, physicians can find it difficult to provide individualized treatment and management strategies for BCa patients. Nevertheless, to date, these two scores are the most common and best-validated models for predicting BCa prognosis. In late 2021, EAU presented a restructured risk group that successfully stratified progression risks in NMIBC [8]. Despite the limitation of overestimating progression in patients with BCG treatment, this new risk stratification is worth looking forward to.

Analyses of genomic alterations in NMIBC revealed complex genomic patterns underlying bladder carcinogenesis; however, the pathological parameters of the tumor do not fully reflect the “intrinsic characteristics” [12,15,17,18]. Earlier studies on BCa performed transcriptomic analyses to develop molecular classification systems for NMIBC [9,12,13,19,20]. The UROMOL project is the most well-known study of NMIBC; the aim was to predict the disease course of BCa using risk scores that combine molecular and clinical risk factors. The UROMOL classification was developed in 2016 as it was based on data from 460 NMIBC patients. The UROMOL 2016 classification system has three gene expression-based classes: classes 1–3. Each class has a different clinical outcome and molecular characteristics [9]. A more recent study reported an integrative multi-omics analysis of NMIBC from a total of 834 patients included in the UROMOL project, and identified four classes (1, 2a, 2b and 3) that reflect tumor biology and disease aggressiveness [12]. Although previous studies identified the molecular characteristics of NMIBC, they provided limited information concerning the association between molecular taxonomy and therapeutic relevance (particularly responses to BCG treatment). Previously, we developed and validated a MSP888 predictor for classifying prognostic subtypes of NMIBC through gene expression profiling of multiple cohorts comprising 948 NMIBC patients [13]. Three subtypes of NMIBC exhibited distinct prognostic features: (1) DP.BCG+ was associated with progression and a positive responsive to BCG; (2) REC.BCG+ was associated with worse RFS and a better response to BCG; and (3) EP exhibited equivocal prognostic behavior.

Intravesical BCG therapy is the most recommended option for intermediate- and high-risk NMIBC patients after transurethral resection of bladder tumor (TURBT) to reduce the risk of NMIBC recurrence and progression [1,3,21]. With BCG treatment, approximately 35% of patients could experience long-term, sustained remissions, although half of patients will suffer recurrence within two years, and about one-third will progress to muscle-invasive stages ultimately requiring radical cystectomy to remove the bladder [3,21]. Risk stratification usually relies on the clinicopathologic features of the disease to provide an ideal therapy for each individual patient [1,3,21]. However, this stratification often underestimates those patients assessed as low-risk NMIBC who were not recommend the BCG treatment. The right risk estimation leads to the right disease management, which directly correlates with the patient’s prognosis and their overall survival. In order to more effectively tailor an appropriate treatment regimen for these low- or intermediate-low-risk patients, a more reliable further stratification is needed [21].

In the current study, we compared the predictive value of the MSP888 with that of the EORTC, CUETO and 2021 EAU risk scores for NMIBC recurrence or progression in a CBNUH cohort (*n* = 178). Furthermore, we compared the predictive value of the MSP888, UROMOL classification, Lund subtype and EORTC risk score for NMIBC progression in the UROMOL cohort (*n* = 460). The results from the CBNUH cohort demonstrated that the MSP888 classifier is an independent risk factor for NMIBC prognosis, and that its performance is superior to that of the EORTC, CUETO and 2021 EAU risk scores, especially for NMIBC patients not treated with BCG (Table 1, Table 2 and Table 3, Figure 2). In the UROMOL cohort, the ability of the MSP888 classifier to predict NMIBC progression was comparable with that of the EORTC score, as well as with the UROMOL classification and Lund subtype (Table 5, Figure 3). In particular, the MSP888 classifier and Lund subtype were independent risk factors for NMIBC progression in patients not treated with BCG (Table 6, Figure 4). Kaplan–Meier survival plots revealed that all four risk estimators (MSP888, UROMOL classification, Lund subtype and the EORTC score) could discriminate NMIBC patients with progression from those without progression (log-rank test, *p* < 0.05 respectively); however, the EORTC result was equivocal. The EORTC scores divided patients into two groups, which is in contrast to the common division into four groups; in addition, the number of tumors (an index used to calculate the EORTC score) was not provided for the UROMOL cohort. Accordingly, these results suggest that our molecular signature-based risk predictor MSP888 could provide a more accurate risk assessment than clinicopathological-based risk scores for those patients not treated with BCG, most of whom were evaluated as low- or intermediate-low-risk NMIBC.

The current study is a second phase validation of the MSP888 predictor for NMIBC patients. However, the small sample size of RNA-seq is the limitation of the present study. Furthermore, a series of tests will be continuously arranged to verify the predictive power of MSP888. Finally, we aimed to manufacture a prognostic molecular subtype-based panel based on counsel to NMIBC patients about their risk of recurrence or progression, followed by the suggestion of individual treatment strategies. In the next generation of risk classification, we are likely to see the inclusion of molecular subtyping with specific treatment considerations.

## 4. Materials and Methods

### 4.1. Study Design

Overall study design is shown in Figure 5. To determine the prognostic utility of the signature-based classifier MSP888, RNA-seq data from 49 NMIBC patients registered at CBNUH were used to establish the MSP888 classifier. The performance of the MSP888 classifier for predicting NMIBC recurrence and progression in 178 NMIBC patients from CBNUH (including 49 patients with newly analyzed RNA-seq data and a published cohort of 129 that was used to develop the subtype classifier and to validate its prognostic value [13]) was compared to that of EORTC, CUETO and 2021 EAU risk groups. In addition, the prognostic ability of the MSP888 classifier, EORTC, the UROMOL classification and Lund subtype were evaluated in 460 NMIBC patients from the UROMOL cohort (a European multicenter prospective study) [9]. In particular, the risk for patients with and without BCG treatment in both the CBNUH and UROMOL cohorts was estimated.

### 4.2. Patients and Tissue Samples

The study methods conformed with the standards set out by the Declaration of Helsinki. Table 7 shows the baseline characteristics of the NMIBC subjects used for RNA-seq analysis. Forty-nine NMIBC patients with distinct clinical outcomes were selected. Tumor tissues were collected from surgically-resected NMIBC from September 2001 to March 2018. All tumors were macro-dissected, typically within 15 min of surgical resection. Each NMIBC specimen was confirmed by pathological analysis of a part of a fresh-frozen specimen obtained from TURBT, and was histologically verified as transitional cell carcinoma. The biospecimens were obtained from the CBNUH (Cheongju, Republic of Korea), which is involved in the National Biobank of Korea. The study was approved by the Institutional Review Board at CBNUH (GR2010-12-010 and GR2020-07-018), and the experiments were undertaken with the informed written consent of all participants. To reduce the chances of confounding factors affecting the analyses, patients diagnosed with concomitant CIS, or CIS lesions alone, were excluded. Tumors were staged (2017 TNM Classification) and graded (1997 WHO Classification) according to standard criteria [3]. The EORTC and CUETO risk scores for both recurrence and progression were calculated [5,6], and patients were stratified into four risk groups according to the calculated scores. Each patient was followed up and managed according to standard guideline recommendations [3]. Surveillance was performed by cystoscopy and upper urinary tract imaging in accordance with European Association of Urology guidelines [3]. In the current investigation, disease recurrence was defined as relapse of primary NMIBC of the same pathologic stage, and progression of NMIBC was defined as TNM stage progression after disease recurrence [22]. The mean follow-up period for NMIBC patients was 64.43 months (range, 4.90–227.33).

### 4.3. RNA Extraction

Total RNA was extracted from tissues using TRIzol reagent (Invitrogen, Carlsbad, CA, USA), as described previously [23], and stored at −80 ℃ until use.

### 4.4. RNA-Sequence Analysis

The value of RNA Integrity Number (RIN) and the DV200 metric were measured using an RNA 6000 Nano Kit and an Agilent 2100 Bioanalyzer (Agilent Technologies, Santa Clara, CA, USA) to confirm the quality and integrity of the RNA. RNA samples with a RIN higher than seven were designated as “good total RNA quality” and selected for downstream application. Total RNA (500 ng) was processed to prepare a whole transcriptome sequencing library. Whole transcriptome RNA was enriched by depleting ribosomal RNA (rRNA) prior to generating the whole transcriptome sequencing library using the MGIEasy RNA Directional Library Prep Kit (MGI Tech Co., Ltd., Shenzhen, China). After rRNA was depleted, the remaining RNA was fragmented into small pieces using divalent cations under elevated temperature. The cleaved RNA fragments were copied to generate first strand cDNA using reverse transcriptase and random primers. Strand-specificity was achieved using an RT directional buffer. This was followed by second strand cDNA synthesis. The cDNA fragments had a single “A” base added prior to ligation of the adapter. The products were then purified and enriched by PCR to create the final cDNA library. The double-stranded library was quantified using a QauntiFluor ONE dsDNA System (Promega, Madison, WI, USA) and add TE buffer to equal 330 ng in a total volume of 60 ㎕. The library was cyclized at 37°C for 60 min and then digested at 37°C for 30 min, followed by cleanup of the circularization product. To generate a DNA nanoball (DNB), the library was incubated at 30°C for 25 min with DNB enzyme. Finally, the library was quantified using a QauntiFluor ssDNA System (Promega, Madison, WI, USA). Sequencing of the prepared DNB was conducted using the MGIseq system (MGI Tech Co., Ltd., Shenzhen, China) with 150 bp paired-end reads. Reference genome sequence data from Homo sapiens were obtained from the NCBI Genome database (assembly ID: GRCh38). Reference genome indexing and read mapping of tissue samples were performed using STAR software (ver. 2.5.4b) [24].

### 4.5. Transcriptomic Summarizing

All of the gene expression data were log2-transformed and quantile-normalized in the current study. The read counts per million fragments mapped value of each sample were measured in the 49 NMIBC patients of the CBNUH-RNAseq cohort. Mean sequencing coverage of RNA-seq data was 47.27× ± standard deviation 16.39× (ranges from 32.70× to 95.10×) and the average mapping rate to reference genome is 93.07% ± standard deviation 1.46% (ranges from 88.86 to 95.20%).

### 4.6. Establishment of MSP888 Clusters in the CBNUH RNA-seq Cohort

In a previous study [13], we generated the MSP888 prediction model by adopting a DBN [14] algorithm. MSP888 is a model incorporating 888 differentially expressed genes among three subtypes according to two sample t-tests. Then, 5 hidden layers were set up, in which 600, 300, 100, 300 and 600 nodes were allocated respectively to construct a fully connected neural network. A Tanh function was used as an activation function for the neural network, and a training procedure was repeated during 1000 epochs. To prevent overfitting or non-convergence of the classifier, the prediction model was pre-trained by an AutoEncoder [14] algorithm. Development of the prediction model was undertaken using the H2O (https://www.h2o.ai (accessed on 3 December 2021)) deep learning platform (ver. 3.32.0.2). Using this training model, evaluation of predicted outcomes in the CBNUH RNA-seq cohort was carried out based on gene expression signatures.

### 4.7. Public Datasets of NMIBC Patients

An RNA-seq dataset from the European UROMOL consortium [9] was downloaded from the ArrayExpress database under accession number E-MTAB-4321. In this dataset, 460 NMIBC samples were used as a validation cohort (the UROMOL cohort). All transcriptomic data used in this study include patient survival and follow-up time to estimate the prognostic relevance of the signatures.

### 4.8. Statistical Analyses

The relationship between each risk predictor and NMBC prognosis was analyzed using Pearson’s Chi-squared test. The interval LR for each stratum was calculated as the likelihood of that test result in patients with a positive test divided by the likelihood of that result in patients with a negative test; this was done to show how likely it is that a patient will experience recurrence or progression. The significance of various clinicopathological variables and the risk estimators was evaluated using univariate and multivariate Cox proportional hazard regression models. Relative risk was determined by calculating hazard ratios (HRs) and 95% confidence intervals (CIs). Kaplan–Meier survival curves were plotted to examine the prognostic value of each risk estimator and compared using the log-rank test. Statistical analyses were performed using IBM SPSS Statistics ver. 24.0 (IBM Co., Armonk, NY, USA) and MedCalc ver. 18.2.1 (MedCalc Software, Mariakerke, Belgium). *p*-Values < 0.05 were considered significant.

## 5. Conclusions

In conclusion, the current study validated the prognostic power of the MSP888 classifier and compared it with that of the EORTC, CUETO and 2021 EAU risk scores. The MSP888 classifier showed superior value for predicting NMIBC prognosis, suggesting that molecular classification-based systems are more accurate than the risk scores based on clinicopathological characteristics. This explains the fact that, compared to the traditional classification system, molecular subtyping reflects the intrinsic characteristics of the tumors, and predicts the prognosis and treatment response of NMIBCs.

## Figures and Tables

**Figure 1 ijms-23-14481-f001:**
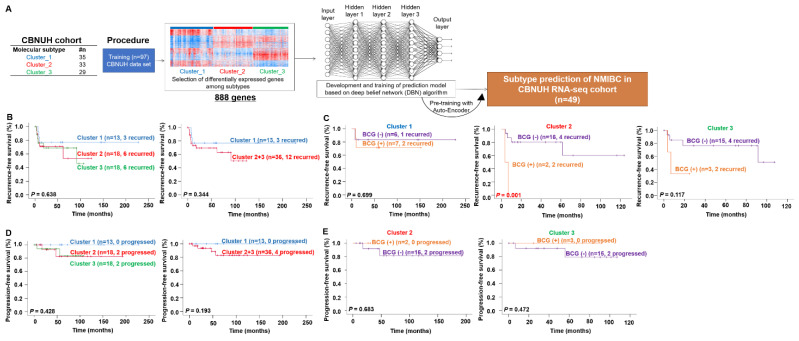
Construction of the MSP888 classifier and validation of its prognostic value in a CBNUH RNA-seq cohort. (**A**) Validation strategy used to construct a prediction model based on the CBNUH RNA-seq cohort. Recurrence-free survival (**B**) and progression-free survival (**D**) of NMIBC patients are predicted by different clusters. The ability of the MSP888 classifier to predict recurrence (**C**) or progression (**D**) after BCG treatment in three subgroups. Patients in Cluster 2 (**E**) derived significant benefit from BCG treatment. Data were plotted according to whether patients received BCG therapy. BCG, Bacillus Calmette-Guérin; CBNUH, Chungbuk National University Hospital; MSP888, a molecular signature-based subtype predictor comprising Cluster 1 (EP, equivocal prognosis), Cluster 2 (REC.BCG+, related to recurrence and response to BCG treatment), and Cluster 3 (DP.BCG+, related to progression and response to BCG treatment).

**Figure 2 ijms-23-14481-f002:**
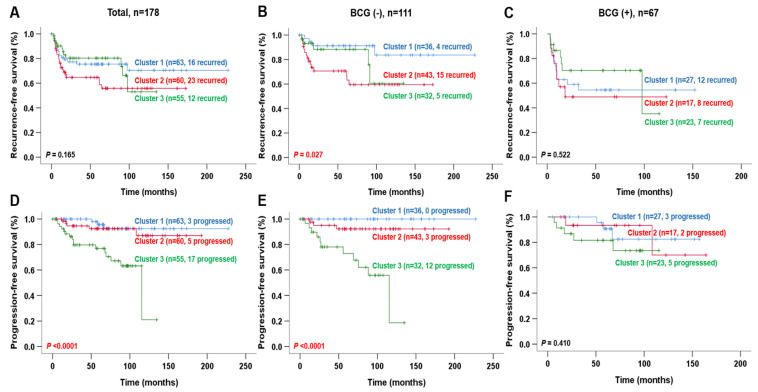
Kaplan–Meier survival curves for risk of recurrence and progression in the CBNUH cohort (based on MSP888 classifier). (**A**–**C**) Recurrence-free survival according to MSP888 classifier. (**D**–**F**) Progression-free survival according to MSP888 classifier. (**A**,**D**), all patients; (**B**,**E**), patients without BCG treatment; (**C**,**F**), patients with BCG treatment. BCG, Bacillus Calmette-Guérin; CBNUH, Chungbuk National University Hospital; MSP888, a molecular signature-based subtype predictor comprising Cluster 1 (EP, equivocal prognosis), Cluster 2 (REC.BCG+, related to recurrence and response to BCG treatment) and Cluster 3 (DP.BCG+, related to progression and response to BCG treatment).

**Figure 3 ijms-23-14481-f003:**
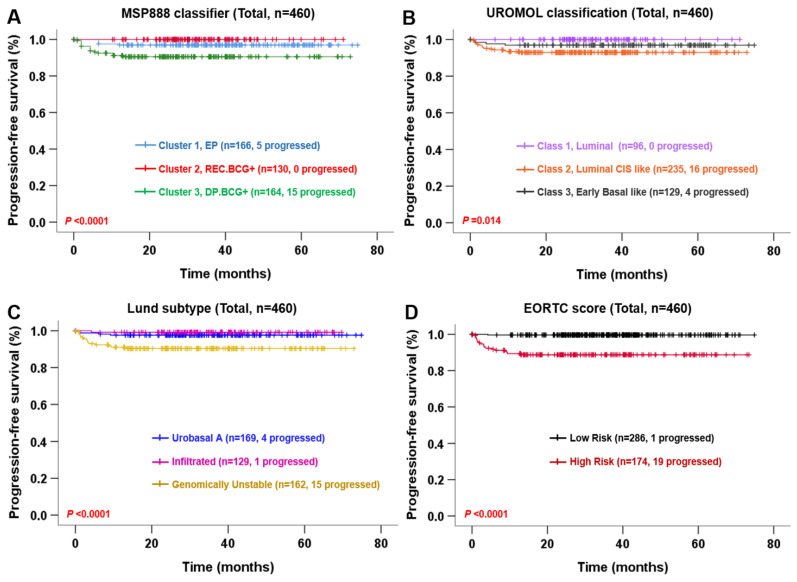
Kaplan–Meier survival curves showing risk of progression in the UROMOL cohort. Progression-free survival according to MSP888 classifier (**A**), UROMOL classification (**B**), Lund subtype (**C**) and EORTC risk score (**D**).

**Figure 4 ijms-23-14481-f004:**
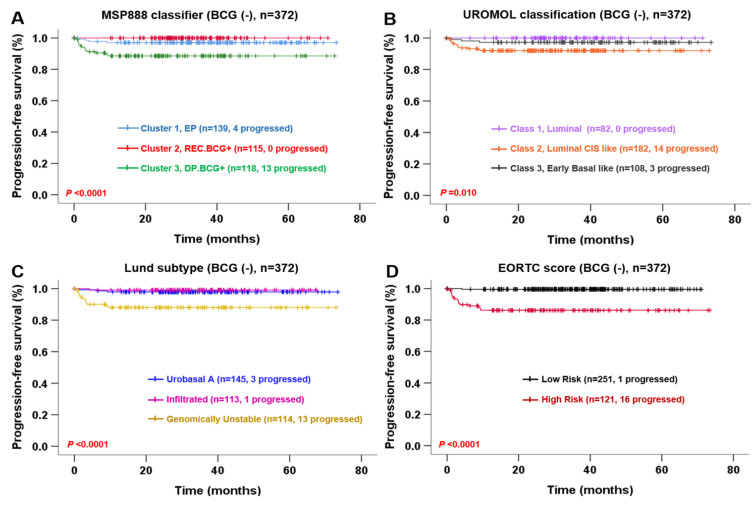
Kaplan–Meier survival curves showing risk of progression in patients in the UROMOL cohort not treated with BCG. Progression-free survival according to MSP888 classifier (**A**), UROMOL classification (**B**), Lund subtype (**C**) and EORTC risk score (**D**).

**Figure 5 ijms-23-14481-f005:**
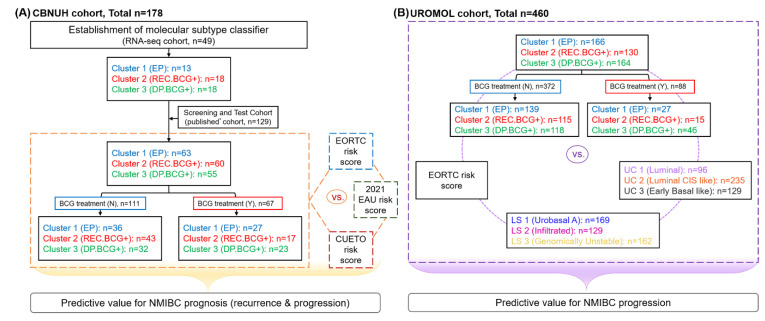
Overall study design. (**A**). Comparison of MSP888 clusters with the EORTC, CUETO and 2021 EAU risk scores for predicting NMIBC prognosis (both recurrence and progression) in the CBNUH cohort. (**B**). Comparison of the MSP888 clusters with the EORTC risk score, the UROMOL classifications and Lund subtypes for predicting NMIBC progression in the UROMOL cohort (a European multicenter prospective study). CBNUH, Chungbuk National University; LS, Lund subtype; NMIBC, non-muscle-invasive bladder cancer; UC, UROMOL classification.

**Table 1 ijms-23-14481-t001:** Pearson’s Chi-Square test for comparison of different risk estimators in the CBNUH cohort.

Group			Total, n = 178				BCG Treatment (−), *n* = 111				BCG Treatment (+), *n* = 67			
			Recurrence (*n*)		Progression (*n*)		Recurrence (*n*)		Progression (*n*)		Recurrence (*n*)		Progression (*n*)	
			No	Yes	No	Yes	No	Yes	No	Yes	No	Yes	No	Yes
MSP888 classifier	1		47	16	60	3	32	4	36	0	15	12	24	3
	2		37	23	55	5	28	15	40	3	9	8	15	2
	3		43	12	38	17	27	5	20	12	16	7	18	5
	Pearson’s Chi-Square *p*-Value		0.115		<0.0001 *		0.024 *		<0.0001 *		0.485		0.526	
	Interval LR (95% CI)	Cluster 1	−		0.306 (0.104–0.901)		0.453 (0.178–1.155)		0.000 (0.000–1.373)		−		−	
		Cluster 2			0.556 (0.247–1.253)		1.942 (1.257–3.000)		0.480 (0.170–1.357)					
		Cluster 3			2.738 (1.863–4.024)		0.671 (0.290–1.556)		3.840 (2.412–6.112)					
EORTC score	1		10	1	17	2	10	0	14	2	0	1	3	0
	2		46	22	53	7	35	13	34	5	11	9	19	2
	3		61	26	78	16	34	11	46	8	27	15	32	8
	4		10	2	5	0	8	0	2	0	2	2	3	0
	Pearson’s Chi-Square *p*-Value		0.331		0.576		0.111		0.936		0.534		0.504	
CUETO score	1		86	27	76	7	56	12	53	3	27	15	23	4
	2		26	17	38	8	20	10	19	5	6	7	19	3
	3		16	6	25	4	10	2	16	2	6	4	9	2
	4		2	1	14	6	1	0	8	5	1	1	6	1
	Pearson’s Chi-Square *p*-Value		0.327		0.078		0.319		0.010 *		0.697		0.989	
	Interval LR (95% CI)	1	−		−		−		0.362 (0.130–1.013)		−		−	
		2							1.684 (0.741–3.829)					
		3							0.800 (0.204–3.134)					
		4							4.000 (1.507–10.614)					
2021 EAU risk group	1		−	−	39	2	−	−	30	1	−	−	9	1
	2		−	−	46	8	−	−	24	3	−	−	22	5
	3		−	−	65	15	−	−	41	11	−	−	24	4
	4		−	−	3	0	−	−	1	0	−	−	2	0
	Pearson’s Chi-Square *p*-Value		−		−		−		0.127		−		0.843	

Cluster 1 = EP, equivocal prognosis; Cluster 2 = REC.BCG+, related to recurrence and response to BCG treatment; Cluster 3 = DP.BCG+, related to progression and response to BCG treatment. EORTC and CUETO score 1 = low risk; 2 = intermediate low risk; 3 = intermediate high risk; 4 = high risk. 2021 EAU risk group 1 = low risk; 2 = intermediate risk; 3 = high risk; 4 = very high risk. CBNUH, Chungbuk National University Hospital; CI, confidence interval; LR, likelihood ratio. * *p* < 0.05

**Table 2 ijms-23-14481-t002:** Univariate and multivariate Cox regression analyses to predict recurrence of NMIBC patients in the CBNUH cohort not treated with BCG.

Variable (*n* = 111)	Univariate Cox Analysis		Multivariate Cox Analysis	
	HR (95% CI)	*p*-Value	HR (95% CI)	*p*-Value
Age ≤70 (Ref.) vs. >70	1.160 (0.659–2.041)	0.607		
Gender Male (Ref.) vs. Female	1.213 (0.451–3.262)	0.702		
Tumor size ≤3 cm (Ref.) vs. >3 cm	0.563 (0.240–1.317)	0.185		
Multiplicity Single 2–7 >8	Ref. 1.427 (0.572–3.555) 1.389 (0.394–4.901)	0.446 0.610		
Stage Ta (Ref.) vs. T1	1.002 (0.444–2.259)	0.996		
1997 WHO Grade 1 2 3	Ref. 1.132 (0.363–3.533) 4.014 (1.467–10.986)	0.831 0.007 *	Ref. 0.663 (0.133–3.304) 1.326 (0.184–9.560)	0.616 0.780
2004 WHO Grade Low (Ref.) vs. High	3.851 (1.622–9.147)	0.002 *	4.469 (0.801–24.929)	0.088
EORTC score Low risk Intermediate low risk Intermediate high risk High risk	Ref. 31744.902 (0.000–8.871 E + 82) 29417.735 (0.000–8.222 E + 82) 1.009 (0.000–2.246 E + 137)	0.910 0.911 1.000		
CUETO score Low risk Intermediate low risk Intermediate high risk High risk	Ref. 3.330 (1.401–7.914) 1.711 (0.376–7.793) 0.000 (0.000)	0.006 * 0.487 0.986		
MSP888 classifier Cluster 1 Cluster 2 Cluster 3	Ref. 3.914 (1.297–11.814) 2.010 (0.538–7.517)	0.015 * 0.299	Ref. 4.104 (1.319–12.770) 1.051 (0.265–4.159)	0.015 * 0.944

Clusters 1, 2 and 3 correspond to the EP, REC.BCG+ or DP.BCG+ subtypes, respectively. * *p* < 0.05. CBNUH, Chungbuk National University Hospital; CI, confidence interval; HR, hazard ratio; NMIBC, non-muscle-invasive bladder cancer; Ref., reference.

**Table 3 ijms-23-14481-t003:** Univariate and multivariate Cox regression analyses to predict progression of NMIBC patients in the CBNUH cohort.

	Total, *n* = 178				BCG Treatment (−), *n* = 111			
Variable	Univariate Cox Analysis		Multivariate Cox Analysis		Univariate Cox Analysis		Multivariate Cox Analysis	
	HR (95% CI)	*p*-Value	HR (95% CI)	*p*-Value	HR (95% CI)	*p*-Value	HR (95% CI)	*p*-Value
Age ≤70 (Ref.) vs. >70								
	2.938 (1.326–6.510)	0.008 *	1.798 (0.800–4.041)	0.156	6.894 (2.151–22.096)	0.001 *	3.801 (1.163–12.420)	0.027 *
Gender Male (Ref.) vs. Female								
	1.070 (0.401–2.853)	0.893			1.689 (0.537–5.312)	0.370		
Tumor size ≤ 3 cm (Ref.) vs. > 3 cm								
	1.163 (0.529–2.557)	0.706			1.239 (0.447–3.434)	0.680		
Multiplicity Single 2–7 >8								
	Ref.				Ref.			
	1.475 (0.618–3.518)	0.381			1.220 (0.375–4.107)	0.741		
	1.950 (0.621–6.124)	0.253			1.750 (0.367–8.340)	0.483		
Stage Ta (Ref.) vs. T1								
	1.154 (0.498–2.675)	0.739			1.164 (0.414–3.273)	0.774		
1997 WHO Grade 1 2 3								
	Ref.		Ref.		Ref.		Ref.	
	4.710 (1.049–21.150)	0.043 *	2.706 (0.548–13.373)	0.222	6.114 (0.709–52.754)	0.100	1.618 (0.097–27.040)	0.738
	10.529 (2.311–47.971)	0.002 *	2.656 (0.381–18.510)	0.324	17.608 (2.204–140.679)	0.007 *	2.016 (0.098–41.672)	0.650
2004 WHO Grade Low (Ref.) vs. High								
	3.926 (1.721–8.957)	0.001 *	1.495 (0.470–4.754)	0.495	10.956 (2.446–49.078)	0.002 *	2.683 (0.291–24.710)	0.384
EORTC score Low risk Intermediate low risk Intermediate high risk High risk								
	Ref.				Ref.			
	1.488 (0.307–7.198)	0.622			1.397 (0.269–7.271)	0.691		
	2.021 (0.482–9.173)	0.323			1.583 (0.342–7.669)	0.544		
	0.000 (0.000)	0.982			0.000 (0.000)	0.988		
CUETO score Low risk Intermediate low risk Intermediate high risk High risk								
	Ref.				Ref.			
	2.503 (0.905–6.921)	0.077			5.314 (1.259–22.426)	0.023 *		
	2.668 (0.773–9.213)	0.121			3.529 (0.579–21.501)	0.171		
	6.596 (2.201–19.764)	0.001 *			14.119 (3.340–59.682)	<0.0001 *		
2021 EAU risk group Low risk Intermediate risk High risk Very high risk								
	Ref.				Ref.			
	4.249 (0.898–20.106)	0.068			4.849 (0.502–46.815)	0.172		
	5.717 (1.299–25.172)	0.021 *			10.134 (1.296–79.261)	0.027 *		
	0.000 (0.000)	0.988			−	−		
MSP888 Classifier Cluster 1&2 (Ref.) vs. Cluster3								
	6.689 (2.817–15.885)	<0.0001 *	4.285 (1.708–10.750)	0.002 *	14.619 (4.042–52.869)	<0.0001 *	7.411 (1.985–27.669)	0.003 *

Clusters 1, 2 and 3 correspond to the EP, REC.BCG+ or DP.BCG+ subtypes, respectively. * *p* < 0.05. CBNUH, Chungbuk National University Hospital; CI, confidence interval; HR, hazard ratio; NMIBC, non-muscle-invasive bladder cancer; Ref., reference.

**Table 4 ijms-23-14481-t004:** Pearson’s Chi-Square test for comparison of the different risk estimators in the UROMOL cohort.

Group			Total, *n* = 460		BCG Treatment (−), *n* = 372		BCG Treatment (+), *n* = 88	
			Progression (*n*)		Progression (*n*)		Progression (*n*)	
			No	Yes	No	Yes	No	Yes
UROMOL classification	Class 1 (Luminal)		95	1	81	1	14	0
	Class 2 (Luminal CIS-Like)		210	25	160	22	50	3
	Class 3 (Early Basal-Like)		124	5	104	4	20	1
	Pearson’s Chi-square		0.002 *		0.002 *		0.663	
	Interval LR (95% CI)	Class 1	0.146 (0.021–1.010)		0.157 (0.023–1.083)		−	
		Class 2	1.647 (1.352–2.008)		1.767 (1.428–2.186)			
		Class 3	0.558 (0.247–1.262)		0.490 (0.195–1.227)			
Lund subtype	Urobasal A		165	4	142	3	23	1
	Infiltrated		123	3	110	3	16	0
	Genomically Unstable		138	24	93	21	45	3
	Pearson’s Chi−Square		<0.0001 *		<0.0001 *		0.579	
	Interval LR (95% CI)	Urobasal A	0.335 (0.133–0.844)		0.268 (0.092–0.784)		−	
		Infiltrated	0.329 (0.111–0.976)		0.351 (0.119–1.030)			
		Genomically Unstable	2.407 (1.903–3.043)		2.902 (2.224–3.787)			
EORTC score	Low Risk, Score ≤6		282	4	247	4	35	0
	High Risk, Score >6		147	27	98	23	49	4
	Pearson’s Chi-square		<0.0001 *		<0.0001 *		0.096	
	Interval LR (95% CI)	Low Risk	0.196 (0.079–0.491)		0.207 (0.084–0.513)		−	
		High Risk	2.542 (2.105–3.069)		2.986 (2.374–3.754)			
MSP888 classifier	Cluster 1		158	8	132	7	26	1
	Cluster 2		130	0	115	0	15	0
	Cluster 3		141	23	98	20	43	3
	Pearson Chi−square		<0.0001 *		<0.0001 *		0.556	
	Interval LR (95% CI)	Cluster 1	0.701 (0.381–1.289)		0.676 (0.353–1.298)		−	
		Cluster 2	0.000 (0.000–0.835)		0.000 (0.000–0.867)			
		Cluster 3	2.257 (1.762–2.892)		2.623 (1.984–3.467)			

Cluster 1 = EP, equivocal prognosis; Cluster 2 = REC.BCG+, related to recurrence and response to BCG treatment; Cluster 3 = DP.BCG+, related to progression and response to BCG treatment. LR, likelihood ratio. * *p* < 0.05.

**Table 5 ijms-23-14481-t005:** Univariate and multivariate Cox regression analysis for predicting progression in the UROMOL cohorts.

Variable (*n* = 460)	Univariate		Multivariate (Age & Stage & MSP888 Classifier)		Multivariate (Age & Stage & UROMOL Classification)		Multivariate (Age & Stage & Lund Subtype)	
	HR (95% CI)	*p*-Value	HR (95% CI)	*p*-Value	HR (95% CI)	*p*-Value	HR (95% CI)	*p*-Value
Gender								
Male (Ref.) vs. Female	1.219 (0.545–2.725)	0.630						
Age								
≤70 (Ref.) vs. >70	2.626 (1.236–5.581)	0.012 ^#^	1.975 (0.914–4.268)	0.083	2.073 (0.963–4.466)	0.062	1.900 (0.881–4.097)	0.101
Stage								
Ta	Ref.		Ref.		Ref.		Ref.	
T1	9.138 (4.202–19.873)	<0.0001 ^#^	6.075 (2.674–13.800)	<0.0001 ^#^	6.815 (3.067–15.140)	<0.0001 ^#^	5.882 (2.609–13.260)	<0.0001 ^#^
CIS	0.000 (0.000–4.47 E + 276)	0.978	0.000 (0.000–7.600 E + 306)	0.978	0.000 (0.000–2.294 E + 291)	0.978	0.000 (0.000)	0.978
Grade								
PUNLMP	Ref.							
Low	1829.893 (0.000–8.723 E + 83)	0.937						
High	7927.856 (0.000–3.777 E + 84)	0.925						
Tumor size								
<3 cm	Ref.							
≥3 cm	1.723 (0.754–3.936)	0.197						
Unknown	0.599 (0.204–1.760)	0.351						
EORTC score								
Low risk (≤6) vs. High risk (>6)	12.487 (4.368–35.700)	<0.0001 ^#^						
UROMOL classification								
Class 1 (Luminal)	Ref.				Ref.			
Class 2 (Luminal CIS-Like)	11.081 (1.501–81.793)	0.018 ^#^			5.220 (0.686–39.737)	0.111		
Class 3 (Early Basal-Like)	3.498 (0.408–29.993)	0.253			2.641 (0.306–22.793)	0.377		
Lund subtype								
Urobasal A	Ref.						Ref.	
Infiltrated	0.972 (0.218–4.345)	0.971					0.900 (0.201–4.031)	0.891
Genomically Unstable	6.967 (2.417–20.087)	<0.0001 ^#^					3.605 (1.200–10.826)	0.022 ^#^
MSP888 classifier *								
Cluster 1&2 (Ref.) vs. Cluster3	5.899 (2.635–13.204)	<0.0001 ^#^	3.025 (1.277–7.166)	0.012 ^#^				

* Clusters 1, 2 and 3 correspond to the EP, REC.BCG+ or DP.BCG+ subtypes, respectively. # *p* < 0.05. CI, confidence interval; HR, hazard ratio; NMIBC, non-muscle-invasive bladder cancer; Ref., reference.

**Table 6 ijms-23-14481-t006:** Univariate and multivariate Cox regression analyses for predicting progression of NMIBC in patients in the UROMOL cohorts not treated with BCG treatment.

Variable (*n* = 372)	Univariate		Multivariate (Age & Stage & MSP888 Classifier)		Multivariate (Age & Stage & UROMOL Classification)		Multivariate (Age & Stage & Lund Subtype)	
	HR (95% CI)	*p*-Value	HR (95% CI)	*p*-Value	HR (95% CI)	*p*-Value	HR (95% CI)	*p*-Value
Gender								
Male (Ref.) vs. Female	1.189 (0.503–2.812)	0.649						
Age								
≤70 (Ref.) vs. > 70	2.962 (1.294–6.777)	0.010 ^#^	2.067 (0.882–4.845)	0.095	2.353 (1.013–5.469)	0. 047 ^#^	2.073 (0.886–4.852)	0.093
Stage								
Ta	Ref.		Ref.		Ref.		Ref.	
T1	10.301 (4.610–23.019)	<0.0001 ^#^	6.513 (2.791–15.196)	<0.0001 ^#^	7.559 (3.300–17.315)	<0.0001 ^#^	6.690 (2.908–15.3)	<0.0001 ^#^
CIS	0.000 (0.000)	0.986	0.000 (0.000)	0.980	0.000 (0.000)	0.985	0.000 (0.000)	0.980
Grade								
PUNLMP	Ref.							
Low	1951.243 (0.000–1.166 E + 85)	0.937						
High	9996.236 (0.000–5.967 E + 85)	0.924						
Tumor size								
<3 cm	Ref.							
≥3 cm	1.953 (0.835–4.568)	0.123						
Unknown	0.515 (0.150–1.769)	0.292						
EORTC score								
Low risk (≤6) *vs.* High risk (>6)	14.175 (4.897–41.033)	<0.0001 ^#^						
UROMOL classification								
Class 1 (Luminal)	Ref.				Ref.			
Class 2 (Luminal CIS-Like)	11.039 (1.488–81.910)	0.004 ^#^			5.529 (0.726–42.100)	0.099		
Class 3 (Early Basal-Like)	2.813 (0.314–25.220)	0.355			2.388 (0.265–21.502)	0.437		
Lund subtype								
Urobasal A	Ref.						Ref.	
Infiltrated	1.266 (0.255–6.271)	0.773					1.184 (0.238–5.876)	0.837
Genomically Unstable	10.430 (3.109–34.989)	<0.0001 ^#^					5.962 (1.733–20.509)	0.005 ^#^
MSP888 classifier *								
Cluster 1&2 (Ref.) vs. Cluster3	7.331 (3.094–17.370)	<0.0001 ^#^	3.753 (1.491–9.447)	0.005 ^#^				

* Clusters 1, 2 and 3 correspond to the EP, REC.BCG+ or DP.BCG+ subtypes, respectively. # *p* < 0.05. CI, confidence interval; HR, hazard ratio; NMIBC, non-muscle-invasive bladder cancer; Ref., reference.

**Table 7 ijms-23-14481-t007:** Clinicopathological characteristics of primary non-muscle-invasive bladder cancer patients used to develop the subtype classifier.

Total *n* = 49	Cluster 1 (*n* = 13)	Cluster 2 (*n* = 18)	Cluster 3 (*n* = 18)	*p*-Value
Mean age ± SD	69.85 ± 13.59	65.61 ± 18.57	70.39 ± 11.26	0.853 ^#^
Gender (%)				0.075 ^§^
Male	10 (76.9)	9 (50.0)	15 (83.3)	
Female	3 (23.1)	9 (50.0)	3 (16.7)	
Tumor size (%)				0.346 ^§^
≤3 cm	6 (46.2)	8 (44.4)	12 (66.7)	
>3 cm	7 (53.8)	10 (55.6)	6 (33.3)	
Multiplicity (%)				0.063 ^§^
Single	2 (15.4)	4 (22.2)	8 (44.4)	
2–7	4 (30.8)	11 (61.1)	6 (33.3)	
>7	7 (53.8)	3 (16.7)	4 (22.2)	
Grade, 1997 WHO grading system (%)				0.078 ^§^
1	5 (38.5)	7 (38.9)	4 (22.2)	
2	6 (46.2)	10 (55.6)	6 (33.3)	
3	2 (15.4)	1 (5.6)	8 (44.4)	
Grade, 2004 WHO grading system (%)				0.002 ^§^
Low	10 (76.9)	13 (72.2)	4 (22.2)	
High	3 (23.1)	5 (27.8)	14 (77.8)	
Stage (%)				0.944 ^§^
TaN0M0	7 (53.8)	10 (55.6)	9 (50.0)	
T1N0M0	6 (46.2)	8 (44.4)	9 (50.0)	
BCG treatment (%)				0.015 ^§^
No	6 (46.2)	16 (88.9)	15 (83.3)	
Yes	7 (53.8)	2 (11.1)	3 (16.7)	
Recurrence –no. of patients (%)				0.789 ^§^
No	10 (76.9)	12 (66.7)	12 (66.7)	
Yes	3 (23.1)	6 (33.3)	6 (33.3)	
Progression –no. of patients (%)				0.455 ^§^
No	13 (100.0)	16 (88.9)	16 (88.9)	
Yes	0 (0)	2 (11.1)	2 (11.1)	

BCG, Bacillus Calmette-Guerin; NMIBC, non-muscle-invasive bladder cancer; SD, standard deviation; WHO, World Health Organization. ^#^ *p*-value obtained using the Kruskal–Wallis H test. ^§^
*p*-values obtained using the χ^2^ test.

## Data Availability

The datasets used and/or analyzed in the current study are available from the corresponding authors upon reasonable request.

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
