# Peer review of "Utility of a Molecular Signature for Predicting Recurrence and Progression in Non-Muscle-Invasive Bladder Cancer Patients: Comparison with the EORTC, CUETO and 2021 EAU Risk Groups"

_ijms, 2022, doi:10.3390/ijms232214481_

Round 1

Reviewer 1 Report

The manuscript is very interesting. However, the authors have assumed that readers are well versed with NIMBC, its treatment and score prediction models. Thus, authors need to work hard abstract and introduction, so that readers can follow their results easily. Please find below the concerns that need to be addressed - 

1.     Abstract – Please explain the clinicopathological factors (EORTC, CUETO and 2021 EAU 22 risk scores. What is BCG? Pleas explain Lund and UROMOL subtypes.  The authors need to rewrite the abstract with greater details.

2.     Please write about the world statistics for NIMBC in the introduction. The authors need to introduce the background of NIMBC.

3.     The authors must share the transcriptome summary – the sequencing depth, coverage, alignment rate. Was principle component analysis performed? If yes, please show it.

4.     The authors need to share the gene signatures of each cluster and correlate with their findings.

5.     In Result Section 2.2, the authors use “interval LR”. Please explain the abbreviation and its significance.

6.     Give significance of BCG treatment in NIMBC. Please state the gaps in research to include BCG treatment cohort in your study.

7.     The authors need to clearly explain the terminologies/abbreviations in the beginning. For examples, the  authors can explain the four risk estimators in the introduction section.

8.     Figure 1 needs better resolution.

9.     The authors can discuss in short about the deep belief network-based deep learning method in addition to citing it.

Author Response

Thank you for reviewer’s precise comments. We have revised our paper and the revised points are highlighted in red in the main article.

  1. Abstract has been revised (word count 199).
  2. Epidemiology of NMIBC is included in the introduction part. (line 40-44)
  3. The transcriptome summary is added as a new part in the material and methods: “4.5. Transcriptomic summarizing” in line 423-429. The predicted outcomes (one of the 3 clusters: EP. or REC.BCG+ or DP.BCG+) of RNA-seq data on the foundation of MSP888 prediction model was evaluated via the deep learning platform to compare its prognostic value with EORTC, CUETO and 2021EAU risk scores in the internal and external cohorts. The main purpose actually focused on the comparison of each risk assessments. Thus, we have not performed the PCA analysis.
  4. Gene signatures of each cluster have been demonstrated in the previous paper (Kim, S.-K.; Park, S.-H.; Kim, Y.U.; Byun, Y.J.; Piao, X.-M.; Jeong, P.; Kim, K.; Lee, H.Y.; Seo, S.P.; Kang, H.W.; et al. A Molecular Signature Determines the Prognostic and Therapeutic Subtype of Non-Muscle-Invasive Bladder Cancer Responsive to Intravesical Bacillus Calmette-Guérin Therapy. Int. J. Mol. Sci. 2021, 22, 1450. https://doi.org/10.3390/ijms22031450). This is a validation study of the molecular cluster (MSP888) as a prognostic predictor for NMIBC, along with the comparison of clinicopathological risk assessments and MSP888 for evaluating NMIBC prognosis.
  5. Full term of LR is added in the article. The interval likelihood ratio (LR) was calculated as the probability of an individual test result occurring when recurrence/progression is present divided by the probability of an individual test result occurring when recurrence/progression is absent to demonstrate how likely it is that a patient will experience recurrence or progression. (line113-117). Its significance is also stated in “4.8. statistical analysis” part of the Materials and Methods. à The interval LR for each stratum was calculated as the likelihood of that test result in patients with a positive test divided by the likelihood of that result in patients with a negative test; this was done to show how likely it is that a patient will experience recurrence or progression.
  6. We have added the explanation in the Discussion part. (line 297-309)
  7. We have revised according to the reviewer’s comment. (line 63-67)
  8. We have re-uploaded a renewal version of Figure 1.
  9. We have added the explanation. (line 431-441)

Reviewer 2 Report

In this manuscript, Piao et al., evaluated the utility of different risk assessments by molecular (a molecular signature-based subtype predictor, MSP888) and clinicopathological factors (EORTC, CUETO and 2021 EAU risk scores) in non-muscle-invasive bladder cancer (NMIBC) patients. To conclude, the MSP888 is favorable for stratifying patients to facilitate optimal treatment.

1.    Figure 1 has has a lower resolution than the others and the writing looks blurred and laborious.

2.    Table 1 is too complicated to be a standard three-line table. The Interval LR (95% CI) of EORTC score and 2021 EAU risk group in Table 1 could be removed.

3.    In CBNUH cohorts, only 49 were newly analyzed by the RNA-sequencing. They were further divided into Cluster 1, 2 and 3. In comparison, the sample size is too small, suggesting that the conclusion should be weakened or discuss the limitations of the study due to the small sample size.

Author Response

Thank you for reviewer’s precise comments. We have revised our paper and the revised points are highlighted in red in the main article.

  1. We have re-uploaded a renewal version of Figure 1.
  2. The Interval LR (95% CI) of EORTC score and 2021 EAU risk group in Table 1 have been removed.
  3. We have revised according to the reviewer’s comment. (line 329-339)

Round 2

Reviewer 1 Report

I appreciate that authors have addressed all the raised concerns. I endorse the manuscript for publication.